REGISTERED REPORT PROTOCOL

# Current trends in the application of causal inference methods to pooled longitudinal observational infectious disease studies—A protocol for a methodological systematic review

Heather Hufstedler[ORCID][1]*, Ellicott C. Matthay[2], Sabahat Rahman[ORCID][3], Valentijn M. T. de Jong[ORCID][4], Harlan Campbell[5], Paul Gustafson[5], Thomas Debray[ORCID][4,6], Thomas Jaenisch[1,7,8], Lauren Maxwell[1], Till Bärnighausen[1,9]

1 Heidelberg Institute of Global Health, Heidelberg Medical School, Heidelberg University, Heidelberg, Germany, 2 Department of Epidemiology and Biostatistics, University of California, San Francisco, California, United States of America, 3 University of Massachusetts Medical School, University of Massachusetts, Worchester, Massachusetts, United States of America, 4 Julius Center for Health Sciences and Primary Care, University Medical Centre Utrecht, Utrecht University, Utrecht, Netherlands, 5 Department of Statistics, University of British Columbia, Vancouver, British Columbia, Canada, 6 Cochrane Netherlands, University Medical Centre Utrecht, Utrecht University, Utrecht, Netherlands, 7 Center for Global Health, Colorado School of Public Health, Aurora, Colorado, United States of America, 8 Department of Epidemiology, Colorado School of Public Health, Aurora, Colorado, United States of America, 9 Harvard T H Chan School of Public Health, Harvard University, Boston, Massachusetts, United States of America

* heather.hufstedler@uni-heidelberg.de

## Abstract

## Introduction

Pooling (or combining) and analysing observational, longitudinal data at the individual level facilitates inference through increased sample sizes, allowing for joint estimation of study- and individual-level exposure variables, and better enabling the assessment of rare exposures and diseases. Empirical studies leveraging such methods when randomization is unethical or impractical have grown in the health sciences in recent years. The adoption of so-called "causal" methods to account for both/either measured and/or unmeasured confounders is an important addition to the methodological toolkit for understanding the distribution, progression, and consequences of infectious diseases (IDs) and interventions on IDs. In the face of the Covid-19 pandemic and in the absence of systematic randomization of exposures or interventions, the value of these methods is even more apparent. Yet to our knowledge, no studies have assessed how causal methods involving pooling individual-level, observational, longitudinal data are being applied in ID-related research. In this systematic review, we assess how these methods are used and reported in ID-related research over the last 10 years. Findings will facilitate evaluation of trends of causal methods for ID research and lead to concrete recommendations for how to apply these methods where gaps in methodological rigor are identified.

**Data Availability Statement:** All relevant data from this study will be made available upon study completion.

**Funding:** This project is funded through the RECODID study, which has received funding from the European Union's Horizon 2020 Research and Innovation Programme under Grant Agreement No. 825746 and the Canadian Institutes of Health Research, Institute of Genetics (CIHR-IG) under Grant Agreement N.01886-000.

**Competing interests:** No authors have competing interests.

## Methods and analysis

We will apply MeSH and text terms to identify relevant studies from EBSCO (Academic Search Complete, Business Source Premier, CINAHL, EconLit with Full Text, PsychINFO), EMBASE, PubMed, and Web of Science. Eligible studies are those that apply causal methods to account for confounding when assessing the effects of an intervention or exposure on an ID-related outcome using pooled, individual-level data from 2 or more longitudinal, observational studies. Titles, abstracts, and full-text articles, will be independently screened by two reviewers using Covidence software. Discrepancies will be resolved by a third reviewer. This systematic review protocol has been registered with PROSPERO (CRD42020204104).

## Introduction

The field of medicine has relied heavily on randomized control trials (RCTs) to infer causality. Though considered the gold standard for causal inference, randomization can be unethical or impractical and so cannot always be used to infer causality at the population level. RCTs also often lack external validity [1, 2]—a crucial element for developing evidence-based public health policy. Longitudinal observational research designs offer the opportunity to gather data on a greater number of people over a larger span of time than most longitudinal RCTs. Longitudinal observational studies facilitate the evaluation of interventions where randomization is not ethical, making them invaluable to public health efforts.

In population science, a large sample size is required. A large sample size is also generally required to examine rare exposures/treatments, as is often the case with infectious disease (ID) research. However, conducting single cohort studies large enough to reach such a large sample size can be too expensive and time-consuming. To overcome this problem, scientists often pool data from numerous studies. While the pooling of both aggregate data (AD) and individual patient data (IPD) are valuable, the pooling of IPD can yield more reliable results than AD [3, 4]. Additionally, pooled analyses across diverse cohorts can offer greater variability in exposure and outcome measures, thereby enhancing power and the ability to detect meaningful associations.

In analysing observational data, the famous phrase *'correlation does not imply causation'* has hindered the use of causal language: academic journals and peers alike often discourage the use of causal terminology. And, although there has been much work done in the field of causal inference with observational data over the last decades [5–7], many authors still use terms that skirt the issue, employing terms like 'link' or 'associated with' [7]. However, as some point out, "the proscription against the C-word is harmful to science. . ."[7].

To infer causality with regards to the health effects of exposures/treatments, health researchers have recently adopted methods, some of which originated in economics, political science, and psychology. Growing use of these methods in epidemiology can enhance the internal validity while maintaining the value of an observational cohort's external validity. They do so by improving our ability to control for observed and/or unobserved confounders. These causal methods include but are not limited to instrumental variables (IV), which, in simple terms, 'looks for a randomized experiment embedded in the observational study' [8]; propensity scores (PS), which can be implemented in several ways, including weighting, matching, or subclassification, e.g., to adjust for covariates, allowing the exposed and

unexposed to be more comparable; difference-in-differences (DiD) models are well-suited for pre-/post-interventions or data with shocks in between; and regression models are widely used in medicine to control for observed confounding.

Applications of these methods listed can be seen in the examination of the impact of infectious disease specialist referrals on health outcomes in France where researchers used IV to tackle methodological issues like selection bias and endogeneity [9], and in one study concerned with the effectiveness of a dengue intervention, researchers used propensity score matching to 'match each treated day with one not treated', a difference-in-difference (DiD) model to examine the 'differences between numbers of dengue cases among scaling up phases', and a linear regression model to estimate the effectiveness of the intervention in the presence of associations between sociodemographic factors' [10].

Each of these causal methods has assumptions (or, conditions) which must be satisfied in order for the method to yield reliable results: e.g. for IV, relevance, exclusion restriction, exchangeability, and monotonicity or homogeneity; and for propensity score methods (PS), exchangeability, consistency, and positivity. Some of the assumptions, or conditions, required for these methods are testable. Some of the assumptions, or conditions, required for these methods are testable. Some of them, though, are untestable, and requires one to evaluate the feasibility of them, often relying on prior literature, theory, causal models, or background knowledge. Discussing the testing of testable assumptions and evaluation of untestable assumptions in the published research article allows the reader to better understand the rigor with which a researcher approached the issue(s). Although similar to multi-centre single cohort studies, implementing causal methodologies with IPD from several cohorts involves a slightly different process and can be more complex, particularly when accounting for, e.g., differences in types of variables that are captured and the ways they are measured, more extreme heterogeneity in cohort composition, or missing data [11, 12]. While these and other causal inference methods have been useful to the study of infectious disease, it is not well-understood how often these methods are being used, in what ways, whether they are being applied rigorously, whether there are gaps in the reporting or application of these methods, or how these factors have changed over time. One study has reviewed the application of causal inference methods applied to time-dependent confounders and also examined which questions are being investigated using non-randomized exposure variables in cohort data deriving from RCTs. They found that the most commonly-implemented method was marginal structural models (MSM) with inverse probability of treatment weighting (IPTW), with the most common question type being the effect of concomitant medication [13]. However, to our knowledge, there exists no methodological review of the application and reporting of causal methods to pooled, observational, longitudinal infectious disease-related studies.

In this systematic review, we seek to fill this gap. We will search the literature (EBSCO, EMBASE, PubMed and Web of Science) using a combination of MeSH and text terms. We will look for infectious disease studies that pool data from 2+ studies. Applying modern causal inference methods to pooled, longitudinal, observational data has the potential to offer important insights into the causes and consequences of infectious diseases. In the face of the Covid-19 pandemic and in the absence of systematic randomization of exposures or interventions, the saliency of these methods is even more apparent. In the short-term, we hope that this review can inform those researchers who are currently analysing observational data with the hope of inferring causality. In the long-term we expect that findings from this review will lead to concrete recommendations for the conduct and reporting of causal inference methods applied to pooled, longitudinal, observational data in ID applications.

### Research question

What are the trends in the conduct and reporting of causal inference methods in ID-related studies using longitudinal, observational data pooled at the participant-level from multiple studies?.

### Hypotheses

We hypothesize that the use of novel and modern causal inference methods has increased in observational ID studies over the last 10 years, but that studies will largely fail to report on key aspects of these methods that are necessary to evaluate their application. For example, we expect that reporting on the quantitative evaluation of assumptions for a given casual inference method (e.g. positivity assumption) will be lacking.

## Materials and methods

### Overview

The protocol for this systematic review has been registered with PROSPERO (CRD42020204104). Our study will test our hypotheses by conducting a systematic review and examine recent trends in the conduct and reporting of causal inference methodology in longitudinal, observational studies pooling data from multiple ID cohorts at the individual level. We will define 'recent' as the last 10 years (2009–2019), but will, due to capacity, limit the search to include studies published at three timepoints 2009, 2014, and 2019.

### Data collection procedures

The following databases will be searched using a combination of MeSH and text terms that is tailored for each database (see S1 Table):

1. EBSCO (including Academic Search Complete, Business Source Premier, CINAHL, Econ-Lit, and PsycINFO)

2. EMBASE

3. PubMed

4. Web of Science

We will include studies that 1) used participant-level data, 2) pooled data from longitudinal, observational cohorts in any location; 3) were focused on infectious disease-related outcomes, 4) estimated a causal effect related to a stated causal question (or, what is interpreted by reviewers as a research study motivated by a causal question; see reference to avoidance of causal language in the introduction), 5) are published in the years 2009, 2014, and 2019 (if there is more than one publication date, we will use the electronic publication date), and 6) have full-text accessible through open access, university license, another collaborator on the project, or by requesting the article directly from the authors. We will also include studies that draw data from RCTs if: a) at least one data source pooled in the analysis is drawn from an observational cohort, or b) the study includes only RCTs but at least one exposure or treatment variable analysed was not that which was randomized.

We will exclude studies that exclusively draw data from RCTs to evaluate randomized exposures or treatments. We will also exclude studies using data from a single-centre cohort or multi-centre single-cohort. We will also exclude studies that: 1) do not employ longitudinal data, 2) estimated an effect size that does not correspond to a research question with the goal of inferring causality (e.g. the study is descriptive or focused on prediction), 3) non-human

studies, 4) protocols, reviews, commentaries, corrections, editorial, erratum, and 5) studies focused on description, prediction, or prognostics.

## Variables

We will extract data on the causal inference designs and methods applied in each study, the quality of reporting on the methods which were applied, and study meta-data (sample size, geographic location of data collection, discipline of parent study, health outcome studied, funding source), etc. See S2 Table for full data extraction sheet.

Examples of data to be collected are:

- Did the authors take any approach(es) to account for differences in variable definitions and data quality across individual cohorts (e.g. any stated information about harmonization efforts) or statistical methods (e.g. adopting measurement error methods)?

- How did the authors deal with missing data within and across studies (e.g. multilevel multiple imputation, or separate imputation for each dataset, or complete case analysis)?

- Do the authors report testing any of the assumptions required for the analysis methods they have chosen to pool the data? Which ones?

- What approach(es) did the authors apply to account for clustering and heterogeneity at the cohort or study level (whichever units are pooled across)? Did the authors adopt a one-stage or two-stage approach?

- Did the authors explicitly state and test the assumptions that are required for methods used to account for clustering and heterogeneity?

- Which causal methods were applied to the pooled data to make causal inferences?

- Do the authors explicitly state or report testing any of the assumptions required for the analysis methods they have chosen to deliver causal effects?

- For untestable assumptions (e.g. unmeasured confounding), did the authors do anything to evaluate the plausibility of those assumptions (e.g. negative control exposures or outcomes, quantitative bias analysis)?

- Did the authors discuss heterogeneity of estimated causal effects and the possible impact on the generalizability of research findings?

## Main outcomes

This is a methodological systematic review designed to establish what causal inference methods are used and how they are reported in studies that use longitudinal data from multiple cohorts (such as pooled cohort studies and individual patient data-meta analyses (IPD-MAs)). Expected outcomes of the review are to establish:

1. Causal inference methods applied in studies using data from multiple cohorts (e.g. instrumental variable approaches, including Mendelian randomization; regression discontinuity; interrupted time series; panel fixed effects; difference-in-differences; G-estimation; multiple regression; propensity score matching; inverse probability of treatment weighting; etc.)

2. Approaches to account for heterogeneity and clustering of the outcome by cohort or data source

3. Approaches to account for differences in measured variables, data quality, or missingness across cohorts

4. Approaches to discussion of methods and the motivating factor(s) in their selection

5. Practices regarding testing of any required assumptions for the chosen causal inference method

6. Reporting standards for studies applying causal inference methods to longitudinal data pooled across multiple cohorts

## Analysis plan

Study records will be uploaded to Covidence [14] and deduplicated. Two reviewers will independently conduct the title/abstract screening in Covidence, and discrepancies will be resolved by a third reviewer. For results flagged as meeting inclusion criteria or uncertain if they meet inclusion criteria, all efforts will be made to access the full-text through databases, university access and collaborators' connections, or requesting the articles directly from the authors. Two reviewers will complete the full-text review. Any discrepancies during the full-text review will be resolved by a third reviewer. One author will extract the data using the data extraction form. The screening process will be documented in a PRISMA flow chart.

We will conduct a narrative summary, and present the results in text, tables and figures. We will summarize trends in all variables collected by reporting frequencies over time. For example, we will summarize trends in causal inference methods by tabulating how frequently each method is used over time (2009, 2014, 2019) and by journal discipline (e.g. economics versus public health). We will also evaluate whether the authors presented sufficient detail on the causal inference method employed and the testing of assumptions required for the methods employed. No meta-analysis of the included studies is planned.

## Supporting information

**S1 Checklist.**
(DOCX)

**S1 Table.**
(PDF)

**S2 Table.**
(PDF)

## Acknowledgments

Membership of the ReCoDID Consortium is available at www.recodid.eu.

## Author Contributions

**Conceptualization:** Heather Hufstedler, Ellicott C. Matthay, Harlan Campbell, Thomas Debray, Lauren Maxwell, Till Bärnighausen.

**Data curation:** Heather Hufstedler, Ellicott C. Matthay, Sabahat Rahman, Lauren Maxwell.

**Formal analysis:** Heather Hufstedler, Ellicott C. Matthay, Sabahat Rahman, Till Bärnighausen.

**Funding acquisition:** Thomas Jaenisch, Lauren Maxwell, Till Bärnighausen.

**Investigation:** Heather Hufstedler, Ellicott C. Matthay, Sabahat Rahman, Till Bärnighausen.

**Methodology:** Heather Hufstedler, Ellicott C. Matthay, Valentijn M. T. de Jong, Harlan Campbell, Paul Gustafson, Thomas Debray, Lauren Maxwell, Till Bärnighausen.

**Project administration:** Ellicott C. Matthay, Thomas Jaenisch, Till Bärnighausen.

**Resources:** Till Bärnighausen.

**Supervision:** Ellicott C. Matthay, Thomas Debray, Lauren Maxwell, Till Bärnighausen.

**Validation:** Heather Hufstedler.

**Visualization:** Heather Hufstedler, Ellicott C. Matthay, Lauren Maxwell, Till Bärnighausen.

**Writing – original draft:** Heather Hufstedler, Ellicott C. Matthay, Sabahat Rahman, Thomas Debray, Lauren Maxwell.

**Writing – review & editing:** Heather Hufstedler, Ellicott C. Matthay, Sabahat Rahman, Valentijn M. T. de Jong, Harlan Campbell, Paul Gustafson, Thomas Debray, Thomas Jaenisch, Lauren Maxwell, Till Bärnighausen.

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
