## [Decision Letter · Decision Letter 0]

22 Feb 2021

PONE-D-20-34304

Current Trends in the Application of Causal Inference Methods to Pooled Longitudinal Observational Infectious Disease Studies -- A Protocol for a Methodological Systematic Review

PLOS ONE

Dear Dr. Hufstedler,

Thank you for submitting your manuscript to PLOS ONE. After careful consideration, we feel that it has merit but does not fully meet PLOS ONE’s publication criteria as it currently stands. Therefore, we invite you to submit a revised version of the manuscript that addresses the points raised during the review process.

We look forward to receiving your revised manuscript.

Kind regards,

Tim Mathes

Academic Editor

PLOS ONE

Journal Requirements:

Reviewers' comments:

Reviewer's Responses to Questions

**Comments to the Author**

1. Does the manuscript provide a valid rationale for the proposed study, with clearly identified and justified research questions?

Reviewer #1: Yes

Reviewer #2: Yes

Reviewer #3: Yes

2. Is the protocol technically sound and planned in a manner that will lead to a meaningful outcome and allow testing the stated hypotheses?

Reviewer #1: Yes

Reviewer #2: Yes

Reviewer #3: Yes

3. Is the methodology feasible and described in sufficient detail to allow the work to be replicable?

Reviewer #1: Yes

Reviewer #2: Yes

Reviewer #3: Yes

4. Have the authors described where all data underlying the findings will be made available when the study is complete?

Reviewer #1: Yes

Reviewer #2: No

Reviewer #3: Yes

5. Is the manuscript presented in an intelligible fashion and written in standard English?

Reviewer #1: Yes

Reviewer #2: Yes

Reviewer #3: Yes

6. Review Comments to the Author

You may also provide optional suggestions and comments to authors that they might find helpful in planning their study.

Reviewer #1: The manuscript reports on a protocol for a methodological review on how causal inference methods are used in analyzing pooled longitudinal observational studies in infections diseases. The protocol is complete and clear. The complete search strategy and data extraction forms are attached as supplementary material.

Major comments

1. I cannot make any guess on how many articles will be retrieved, but I thought that limiting the search to three years (2009, 2014, 2019) may yield a limited number of studies included. Are there fallback solutions to extend the search is the number of studies would be relatively small? (I acknowledge that this leaves the definition of a “relatively small” number unanswered).

2. I also wondered whether the search strategy would be specific enough to spare a lot of screening time. I am not a specialist of building search equations, but I can only underline the risks of screening hundreds (thousands?) of articles on tile and abstract to include a small number at the end.

3. I understand specificities of pooled data analyses, in particular regarding the need to account for heterogeneity between cohorts, but to some extent this is not markedly different from heterogeneity between facilities in a multicenter single cohort study (that are excluded), at least from a statistical point-of-view. There may be important differences with e.g. systematically missing data that are not recorded in a specific study that require specific methods (e.g. Resche-Rigon, White. Stat Methods Med Res 2018;27:1634-1649), and the paper also mentions approaches to account .for differences in data quality or definitions. But some more discussion on why focus only on pooled analyses could be given.

Minor comments

1. The meaning of “pooling” (that I understood as pooling different observational studies together) could be clarified in the abstract.

2. Is there really a need for both IV and IVA acronyms? The first one is largely sufficient.

Reviewer #2: This manuscript is a protocol for a methodological systematic review the application of causal inference methods to Pooled longitudinal observational infectious disease studies. No statistical meta-analysis will be needed. This study will be a descriptive summary of the publications that applied causal inference methods. It will be interested in looking at the summary rather than this protocol.

Lines 153-154, why do you select studies published at three timepoints instead of all related publications over the last 15 years?

Reviewer #3: Manuscript ID: PONE-D-20-34304

Title: Current Trends in the Application of Causal Inference Methods to Pooled Longitudinal Observational Infectious Disease Studies -- A Protocol for a Methodological Systematic Review

I thank both the Editor and the Authors for the opportunity to review the above-mentioned manuscript.

In brief, this paper reports the protocol for an ambitious review on causal inference methods applied to pooled observational data in infectious disease studies. The paper promises a useful and most-needed systematic review, which will help researchers and clinicians to better understand the respective usages and pros/cons of the different methods under consideration. While I look forward to the results of this systematic review, I have a couple of very minor comments for this protocol. I hope this will help the Authors clarify some points.

***Abstract:

- Line 39: The term “individual-level effects” can be confusing for causal inference researchers, as it may refer to “individual treatment effects” (i.e. the causal effect of an exposure/intervention in a particular individual; note, this quantity is unmeasurable.) I suppose the Authors refer to the individual-study-level effects instead;

- Line 42: Causal methods can address both/either unmeasured and/or measured confounders;

- Line 44: I understand a big C-something has been invading our lives since last year, but I find it a bit out of context since this systematic review includes studies from 2009, 2014 and 2019 – and not 2020, as I understand.

***Introduction:

- The Authors could add one or two sentences on the different assumptions needed for the causal methods under consideration (e.g. IV methods addressed unmeasured confounders but require specific IV assumptions [relevance, no shared causes with the outcome and exclusion restriction]; PS methods rely on the assumption of no unmeasured confounders and require positivity);

- Line 104: The brief one-sentence summary on PS methods is vague and incorrect: PS methods do not necessarily rely on matching (e.g. weighting, standardization, etc. can be applied too) to provide covariate balance;

- Line 119-120: the sentence “with non-randomized exposure data from RCTs” sounds paradoxical. Could the Authors reformulate?

***Methods:

- Line 166: The Authors state that they will include studies that “4) estimated a causal effect related to a stated causal question”. While it sounds fair, I fear that causal questions are unfortunately too rarely explicitly stated in practice (see Hernan MA. The C-Word: Scientific Euphemisms Do Not Improve Causal Inference From Observational Data. Am J Public Health. 2018;108(5):616-619.)

- Amongst the bullet points for the extraction, I would also be interested in the covariate selection for adjustment (e.g. Did the study explicitly specify that covariates adjusted for were confounders [and not mediators] of the causal relationship?)

***Other comments:

- I found some typos throughout, and some sentences quite long and heavy to read (in particular, in the introduction).

- Some key references about causal inference methods would be appreciated – e.g. Hernan and Robins book or others.

7. PLOS authors have the option to publish the peer review history of their article (what does this mean?). If published, this will include your full peer review and any attached files.

Reviewer #1: No

Reviewer #2: No

Reviewer #3: No

---

## [Author Response · Author response to Decision Letter 0]

24 Mar 2021

Thank you very much for the constructive as well as positive feedback. I believe we have addressed all of the constructive feedback in a letter attached to the submission. As I said in the comment previously, I believe we have addressed everything except for the request to change our citations from parentheses to brackets. We have tried for the better part of two days without success, and have chosen to submit without this change as to not miss the new deadline. Our apologies, and I hope that you can assist us in aligning with your formatting requirements. Otherwise, we hope we have addressed every other suggestion or comment satisfactorily. Thank you.

---

## [Decision Letter · Decision Letter 1]

7 Apr 2021

PONE-D-20-34304R1

Current Trends in the Application of Causal Inference Methods to Pooled Longitudinal Observational Infectious Disease Studies -- A Protocol for a Methodological Systematic Review

PLOS ONE

Dear Dr. Hufstedler,

Thank you for submitting your manuscript to PLOS ONE. After careful consideration, we feel that it has merit but does not fully meet PLOS ONE’s publication criteria as it currently stands. Therefore, we invite you to submit a revised version of the manuscript that addresses the points raised during the review process.

Please check correctness of citation and proofread the manuscript. 

We look forward to receiving your revised manuscript.

Kind regards,

Tim Mathes

Academic Editor

PLOS ONE

Journal Requirements:

Reviewers' comments:

Reviewer's Responses to Questions

**Comments to the Author**

1. Does the manuscript provide a valid rationale for the proposed study, with clearly identified and justified research questions?

Reviewer #2: Yes

Reviewer #3: Yes

2. Is the protocol technically sound and planned in a manner that will lead to a meaningful outcome and allow testing the stated hypotheses?

Reviewer #2: Yes

Reviewer #3: Yes

3. Is the methodology feasible and described in sufficient detail to allow the work to be replicable?

Reviewer #2: Yes

Reviewer #3: Yes

4. Have the authors described where all data underlying the findings will be made available when the study is complete?

Reviewer #2: Yes

Reviewer #3: Yes

5. Is the manuscript presented in an intelligible fashion and written in standard English?

Reviewer #2: Yes

Reviewer #3: Yes

6. Review Comments to the Author

You may also provide optional suggestions and comments to authors that they might find helpful in planning their study.

Reviewer #2: My comments have been addressed.

Reviewer #3: I thank the Authors for addressing my previous comments/suggestions. I enjoyed reading the revised manuscript, which I think reads very well.

I have a very minor remark, which may however have its own importance: something wrong may have happened with the editing of the references in the revised manuscript - some citations have been deleted in the main text (in particular, Introduction). Could the Authors amend this, and double-check/proof-read before submitting?

I have no further comments other than this one. I thank the Authors for their work.

7. PLOS authors have the option to publish the peer review history of their article (what does this mean?). If published, this will include your full peer review and any attached files.

Reviewer #2: No

Reviewer #3: **Yes: **T.L. Nguyen

---

## [Author Response · Author response to Decision Letter 1]

9 Apr 2021

Thank you for the the opportunity to correct and re-submit our protocol. We have added the two citations which were removed in the introduction, and I believe we have caught both instances of the extra periods, commas, or duplicate parentheses and brackets, as well as the 2 instances where PubMed was written as Pubmed. Thank you.

---

## [Editor Report · Decision Letter 2]

14 Apr 2021

Current Trends in the Application of Causal Inference Methods to Pooled Longitudinal Observational Infectious Disease Studies -- A Protocol for a Methodological Systematic Review

PONE-D-20-34304R2

Dear Dr. Hufstedler,

We’re pleased to inform you that your manuscript has been judged scientifically suitable for publication and will be formally accepted for publication once it meets all outstanding technical requirements.

Kind regards,

Tim Mathes

Academic Editor

PLOS ONE
---

## [Editor Report · Acceptance letter]

20 Apr 2021

PONE-D-20-34304R2 

Current Trends in the Application of Causal Inference Methods to Pooled Longitudinal Observational Infectious Disease Studies -- A Protocol for a Methodological Systematic Review 

Dear Dr. Hufstedler:

I'm pleased to inform you that your manuscript has been deemed suitable for publication in PLOS ONE. Congratulations! Your manuscript is now with our production department. 

Kind regards, 

on behalf of

Dr. Tim Mathes 

Academic Editor

PLOS ONE